# Desire for Children and Distress in Women with Hereditary Cancer Syndromes

**DOI:** 10.3390/ijerph192114517

**Published:** 2022-11-04

**Authors:** Anna Maria Kastner, Hatice Kübra Kocak, Josefine Fischer-Jacobs, Andrea Hahne, Tanja Zimmermann

**Affiliations:** 1Department of Psychosomatic Medicine and Psychotherapy, Hannover Medical School, Carl-Neuberg-Straße 1, 30625 Hannover, Germany; 2BRCA-Netzwerk e.V., Thomas-Mann-Street 40, 53111 Bonn, Germany

**Keywords:** psycho-oncology, tumor disposition syndrome, hereditary cancer syndrome, distress, desire to have children

## Abstract

The diagnosis of a hereditary cancer syndrome can be psychologically stressful and influence family planning. This study aimed to gain insights into the relationship between the desire for children and the distress of female carriers. Women (*N* = 255) with different hereditary cancer syndromes were assessed from November 2019 to July 2021 at genetic counseling centers, the centers of the German HBOC-Consortium and the centers of the German HNPCC-Consortium regarding their distress levels with the NCCN Distress Thermometer (DT). The desire for children was measured by self-developed questions. Levels of distress and desire for children were evaluated descriptively. Factors influencing the desire for children and distress were calculated using binary logistic regression: 56% (*n* = 51) of 18- to 39-year-old participants reported a desire to have children; 70.6% of the carriers with a desire for children indicated a need for advice from their physicians regarding family planning. The diagnosis led 61.5% to postpone the timing of family planning, and the majority (68.8%) opted for an earlier birth. Carriers had higher levels of distress. Younger carriers (*p* = 0.037) and those living in poorer economic circumstances (*p* = 0.011) were more distressed. The diagnosis of hereditary cancer syndrome affects family planning. The results emphasize the importance of physicians addressing family planning in their counseling sessions.

## 1. Introduction

Every year, about half a million people in Germany develop cancer [1]. It can be assumed that these numbers will increase due to rising life expectancy and improvements in early cancer detection and treatment [2]. Approximately 10–20% of all cancers have a genetic cause [3]. Nowadays, more than 40 hereditary tumor syndromes are known, to which underlying gene mutations can be assigned [4]. Hereditary Breast and Ovarian Cancer syndrome (HBOC) and Hereditary Non-Polyposis Colon Cancer syndrome (HNPCC) are the most common [5]. Carriers have significantly increased lifetime risk for developing cancer compared with the general population [6,7]. In cancer patients with early age of onset (age < 60 years), familial accumulation of the syndrome-specific cancer spectrum, multiple neoplasms at the same time, or the recurrence of cancer after a short time, the possibility of a hereditary cancer syndrome should be considered [7]. Patients with these characteristics and any relatives should therefore be referred to a clinical geneticist for further counseling, risk assessment, possible testing, and advice on prevention and treatment.

A positive test result can affect life and family planning [8,9] and cause psychological distress. Especially, within the first weeks after receiving a positive test result, carriers experienced increased levels of distress, anxiety, and/or depression [10,11]. The timing of childbearing may be brought forward or postponed, or even eliminated. The restriction of breastfeeding due to the removal of the breast and possible infertility due to the removal of the ovaries can also be stressful. The onset of cancer and the subsequent therapy can also influence the desire to have children: early menopause, sexual dysfunction, and hormonal disorders are possible side effects of hormone and chemotherapy [12]. Carriers often have a family history of cancer loss while having to make decisions at a young age with far-reaching consequences regarding childbearing, prophylactic surgeries, and family communication [13,14]. In addition, factors such as the uncertainty of coverage of fertility preservation measures by health insurance companies and possible financial consequences and effects on life and disability insurance can be a cause for concern. Particularly for women with a desire for children, the knowledge of autosomal dominant inheritance and fertility-restricting interventions (such as chemotherapy and prophylactic surgery) can lead to distress [15]. Psychosocial support for those affected is therefore particularly important.

The present study aims to provide an insight into the relationship between the experience of distress and the desire to have children in women with hereditary cancer syndromes in Germany. To the best of our knowledge, this is the first German study on this matter. While most international studies only addressed women with HBOC, we included women with all hereditary cancer syndromes. In general, there is a knowledge gap regarding the extent to which the topic of childbearing is relevant to carriers and, accordingly, whether and how medical and psychological staff can address possible concerns and provide support.

### Study Objectives

This study investigated the desire for children (1) and amount of distress (2) that German-speaking female carriers for hereditary cancer syndromes experienced. It is hypothesized that mutation carriers have a desire to have children and show increased distress. Furthermore, the aim was to identify influencing factors on the desire for children (3) and on distress (4). Based on current research, it can be assumed that age (younger), education (lower), economic situation (worse), marital status (low), presence of children, previous or current cancer, prophylactic surgery, distress (high) and desire for children can be influencing factors on mental distress. Studies on factors influencing the desire for children are few, so the influencing factors presented above will be exploratory.

## 2. Materials and Methods

### 2.1. Data Sampling

This cross-sectional multi-center study obtained data from German-speaking women with a range of hereditary cancer syndromes. The ethical committee of the Hannover Medical School, Germany, approved the study and all procedures (8541_BO_K_2019). All participants received information in written form before participation and gave written informed consent before taking part in the online survey. Inclusion criteria were a diagnosis of a hereditary cancer syndrome, age of majority, and sufficient knowledge of the German language. Persons below the German legal age of majority (<18 years) and with severe mental diseases were excluded.

Recruitment took place from November 2019 to July 2021 at genetic counseling centers, the centers of the German HBOC-Consortium and the centers of the German HNPCC-Consortium through information material and involved clinicians. Patient self-help groups were involved in recruitment and contacted participants through their websites, letters, e-mail newsletters, and social media channels. Participants confirmed inclusion criteria on the website and then participated in the online survey.

### 2.2. Participants

*N* = 286 persons (*n* = 255 women, *n* = 31 men) participated in the study. The *n* = 31 men were excluded from the current analysis to avoid confounding with gender. Accordingly, the data of *n* = 255 women with hereditary cancer syndromes were examined. The average age was 44.64 years (SD = 11.24, range: 18–77). The mean age for young carriers (<40 years) [16] was 32.78 (SD = 4.51, range = 18–39). The majority of the sample was married or in a relationship (75.3%), had a university degree (39.2%), and lived in good economic circumstances (37.6%). About two thirds (65.1%) of the carriers had children. The majority had either HBOC (76.7%) or HNPCC (12.9%). Furthermore, about half of the participants (53.3%) had taken prophylactic measures. Table 1 contains more detailed information on the sample’s demographics.

### 2.3. Measures

#### 2.3.1. Distress

Distress was assessed with the German version of the *NCCN Distress Thermometer* (DT) [17]. Participants rated their subjectively experienced distress in the previous week on a single 11-point scale ranging from 0 (“no distress”) to 10 (“extreme distress”).

According to international recommendations, a cut-off of ≥5 is interpreted as clinically relevant and in need of support. The DT can be complemented by a problem list, which was not applied in this study. In a German sample, the discriminating power of the DT was particularly good when screening for higher levels of anxiety or/and depression. For a score of 5, sensitivity was 97% and specificity was up to 41%. Due to its high acceptance, its brevity and practice orientation, the DT is recommended as a screening tool to assess psychosocial distress in cancer patients [17].

#### 2.3.2. Desire to Have Children

The desire to have children was assessed using items developed by the study team. Further items on influencing factors were only answered by participants who had indicated a general desire to have (more) children. The first part of the questionnaire consisted of five questions about personal attitudes toward family planning. An example of such a question is “I can imagine having (more) children later.” The items were to be answered dichotomously with the expression “yes” and “no”. In the second part, another six items on concerns about having children were asked on a five-point scale ranging from 1 = “strongly disagree” to 5 = “strongly agree”. These six items were combined by the author to form a “Worries about wanting a child” scale. For the calculation in SPSS, the item “Having children is part of a fulfilled life for me.” was inverted. In the context of this study, the internal consistency of this scale was very good (Cronbach’s α = 0.81; [18]).

### 2.4. Statistical Analysis

Frequencies as well as means and standard deviations were used for descriptive analysis. For the primary regression–analytical evaluation, a sample size of *n* = 118 evaluable participants was required in order to be able to prove the assumed small effect (0.15) with a power of 80% based on a significance level of 0.05 (calculation with G*Power). The questions on childbearing were calculated only with the data of participants between 18 and 39 years of age (*n* = 91), because after 40 years of age fertility of women decreases rapidly [19] and late pregnancy entails risks for mother and child [20]. This was performed in an attempt to counteract bias in the results due to reproductively infertile participants. The evaluation of the distress levels and the questions on childbearing was descriptive. The conditions of a normal distribution were not met. Nevertheless, parametric methods were used, since they are considered robust against violations of the normal distribution with the existing sample size.

To determine influencing factors for the desire to have children as well as distress binary logistic regressions were used. The preconditions for this analysis were fulfilled as follows. The dependent variables are nominally scaled with exactly two values (dichotomous). The question about the desire to have children was answered dichotomously with yes or no. The dependent variable “distress” was dummy-coded: Values between 0 and 4 were, according to Mehnert et al. [17], assigned the value 1 (=not distressed) and values between 5 and 10 were assigned the value 2 (=distressed). The independent variables considered were age, education, economic situation, marital status, presence of children, the desire to have children, cancer, and prophylactic surgery. Correlations among predictors were low (r < 0.70; [21]), indicating that multicollinearity did not confound the analysis. In addition, there were no outliers in the data: no studentized excluded residuals were above the cut-off value of ≥3, the leverage values were below the cut-off value of ≤0.2, and the Cook distances were far from the cut-off criterion of 1. The condition of linearity was checked using the Box–Tidwell procedure. The Bonferroni correction was applied to all terms in the model. Based on this, linearity could be assumed for all variables. A sufficient sample size assuming 10 cases per predictor is given [22].

All statistical analyses were performed in IBM SPSS Statistics 26 (IBM, Armonk, NY, USA) and all tests were based on a significance level of 0.05.

## 3. Results

### 3.1. Desire for Children in Women with Hereditary Cancer Syndromes

More than half of the participants between 18 and 39 years had a desire to have children (*n* = 51, 56%). Of these, 55.8% (*n* = 29) had a current desire for children and 82.4% (*n* = 42) said they could imagine having (more) children in the future. About 70.6% (*n* = 36) expressed the need for advice from their physicians regarding family planning; 9.8% (*n* = 5) were advised by a physician not to become pregnant or to have a child. In addition, 11.8% (*n* = 6) of the participants thought that their social environment would view it critically if they had a (further) child. The majority (61.5%; *n* = 32) reported a change in the timing of family planning due to the hereditary cancer syndrome diagnosis: 68.8% (*n* = 22) intended to have children earlier and 18.8% (*n* = 6) later than originally planned. Some women were still undecided due to the risk of inheritance (*n* = 2, 6.3%). Participants who indicated a desire to have children were on average M = 2.91 (SD = 0.75) concerned about childbearing (scale: 1 = “strongly disagree” to 5 = “strongly agree”). Furthermore, the desire to have children was compared in women with/without children and prophylactic surgery: 40.9% (*n* = 18) of participants with children and 73.3% (*n* = 33) of participants without children had a desire to have children. In addition, 53.3% (*n* = 16) of women with prophylactic surgery and 67.3% (*n* = 35) without prophylactic surgery desired children.

### 3.2. Distress in Women with Hereditary Cancer Syndromes

Women with hereditary cancer syndromes had a mean distress score of M = 6.22 (SD = 2.67). The value was above the cut-off value of ≥5. Thus, the average distress of carriers is to be classified as “clinically conspicuous” [17]. In the categorical evaluation, 62.7% (*n* = 160) mutation carriers were above the cut-off ≥5.

### 3.3. Factors Influencing the Desire to Have Children in Women with Hereditary Cancer Syndromes

The binary logistic regression model was statistically significant χ^2^ (8) = 16.633, *p* = 0.034, with a good variance resolution of Nagelkerke’s *R*^2^ = 0.224 according to Backhaus et al. [23]. The overall percentage of correct classification was 69.2%, with a sensitivity of 80.4% and a specificity of 55.0%. Of the eight predictors included in the model—age, education, economic situation, marital status, presence of children, cancer diagnosis, prophylactic surgery and distress—one predictor was significant. Having cancer had a significant effect on childbearing (*p* = 0.036). Individuals who did not have cancer had a higher desire to have children with an odds ratio of 3.079 (95% CI 1.076–8.810) and a relative probability of 207.9% (3.079 − 1 = 2.079). All model coefficients and odds ratios can be found in Table 2.

### 3.4. Factors Influencing Distress in Women with Hereditary Cancer Syndromes

The binary logistic regression model was statistically significant, χ^2^ (8) = 20.900, *p* = 0.005, with an acceptable variance resolution of Nagelkerke’s *R*^2^ = 0.110, according to the recommendations of Backhaus et al. [23]. In the model, 61.2% of cases were correctly predicted, with a sensitivity of 57.1% and a specificity of 65.1%. Of the eight predictors included in the model—age, education, economic situation, marital status, presence of children, cancer diagnosis, prophylactic surgery and the desire for children—two predictors were significant. Age (*p* = 0.037) and economic situation (*p* = 0.011) significantly predicted psychological distress. As age decreases by one year, psychological distress increases with an odds ratio of 0.970 (95% CI 0.942–0.998) and a probability of 3% (0.970 – 1 = 0.03). The worse the economic situation, the higher the psychological distress, with an odds ratio of 1.441 (95% CI 1.087–1.912) and a relative probability of 44.1% (1.441 – 1 = 0.441). All model coefficients and odds ratios can be found in Table 3.

## 4. Discussion

This study examined German-speaking women with different hereditary cancer syndromes with and without cancer regarding their desire to have children and the amount of distress they experienced. Additionally, the aim was to identify influencing factors for the desire to have children and distress.

More than half (56%) of the 18- to 39-year-old women with hereditary cancer syndromes had a desire for children and were moderately concerned about having children. In comparison, in the last published official statistics, 47% of women between 20 and 40 years of age wished to have (another) child [24]. Thus, the desire to have children among carriers was about as high as in a population-representative sample. Nevertheless, the hereditary cancer syndrome had an impact on the scheduling of family planning in about 61.5%. In the majority of those (68.8%) the diagnosis led to earlier family planning than originally planned. Participants without prophylactic surgeries showed a higher desire to have children than participants who had undergone prophylactic surgery. Fertility limitation due to prophylactic surgeries (23.6% of 18–39-year-old women with prophylactic surgery had an oophorectomy or hysterectomy) may have influenced attitudes toward childbearing. The desire to have children was also influenced by a previous cancer diagnosis. Healthy carriers between 18 and 39 years of age had a higher desire to have children than carriers who have had cancer. This result is not surprising considering the influence of cancer therapy on fertility. Chemotherapy and radiotherapy are common cancer therapy procedures that may involve ovarian insufficiency [25]. In addition, it is recommended to refrain from pregnancy during chemo- and radiotherapy due to a high risk of malformation of the child [26] as well as for at least six months after the end of therapy [27].

In addition, over 60% of mutation carriers had increased levels of psychological distress. In the studies of Lynch et al. [28], den Heijer et al. [29], Reichelt et al. [30], Vajen et al. [31] and Voorwinden and Jaspers [32], women with a known gene mutation also had a higher level of distress. As hypothesized, already diseased mutation carriers showed significantly higher expression in their psychological distress than healthy mutation carriers. However, it cannot be excluded that the increased expression of the mutation carriers with cancer is due to the cancer and not to the gene mutation. Studies on psychological stress in cancer patients also showed increased stress, which was determined in particular by fatigue and sleep problems [33,34]. Distress was influenced by age. The experience of distress decreased with increasing age. Other studies also showed that younger women experienced greater psychological stress [35,36]. Furthermore, the economic situation of the carriers was another influencing factor in distress. The worse the economic situation, the higher was the distress. This result is consistent with the current state of research [37,38]. The presence of children or the desire to have children had no significant effect on distress levels. In contrast, another study concluded that carriers with children were more distressed than those without children [36].

### Limitations

There are a number of limitations to the study that need to be considered. First, the data are based entirely on self-report measures and the study design is cross-sectional; these methods can pose issues that are common in psychological research, namely, response bias and common method variance. Moreover, despite its size, the sample is not necessarily representative in terms of the distribution of hereditary cancer syndromes. The survey was in German. Individuals who could not read and understand German were thus directly excluded. Our study did not distinguish between cis and trans persons; thus, the sample may not accurately reflect the possibility of childbearing. In addition, the questions on childbearing were not standardized. Furthermore, all questions on the desire to have children were only filled out by those participants who indicated to have a desire for children. As a result, the subgroup (18- to 39-year-old carriers with a desire to have children) was small.

## 5. Conclusions

Having children is a reproductive right, and identifying and addressing factors related to this is important especially in vulnerable individuals including women with hereditary cancer syndromes. This study targeted this issue. The majority of the 18- to 39-year-old carriers had a desire for children. A large majority felt the need for medical advice on childbearing. This finding emphasizes the importance of actively addressing reproductive concerns in medical counselling. There should be room for the medical and psychosocial concerns that may arise, such as interfamilial conflicts and complex feelings of guilt, fear, joy, etc. This seems to be especially important because women with a desire for children tend to be young and our data have shown that young carriers experience higher distress. Accordingly, there also seems to be a relationship between the desire for having children and distress in women with hereditary cancer syndromes. Furthermore, for cancer patients who have to decide for or against fertility-preserving and fertility-restricting therapies within a short time after diagnosis, such counseling is essential [39]. Various studies have shown that in women with cancer, the main concern is not only the fear of death, but also the threat of loss of fertility [40].

In summary, this study provided the first insights into the desire to have children and the distress of women with hereditary cancer syndromes. Due to the continuous development of genetic diagnostics, further risk genes for hereditary cancer are constantly being discovered [41]. It can be assumed that, in the future, the number of positive genetic test results will increase due to improved diagnostics and the discovery of new cancer genes. Due to the wide variety of syndromes and manifestations, clinicians from almost every specialty are involved in the care of carriers. Therefore, it seems important to continue to increase awareness of the special situation of carriers, their need for risk education and care adapted to the specific problems associated with the hereditary component that can affect the whole family. The study findings can highlight the reproductive concerns of these women, and inform health professionals to consider these concerns while counselling. Further research on the desire to have children and the experience of distress is necessary to develop and evaluate psychosocial interventions and optimize medical care.

## Figures and Tables

**Table 1 ijerph-19-14517-t001:** Demographic characteristics of the sample.

	Women with Hereditary Cancer Syndromes (*n* = 255)
**Age in years** (*M, SD, range*)	44.64 (11.24; 18–77)
**Relationship status** (*n*, %^1^)	
Single	44 (17.3)
Married/in a relationship	192 (75.3)
Divorced/separated	19 (7.5)
**Education** (*n*, %^1^)	
<10 years	10 (3.9)
>10 years	94 (40.7)
University degree	100 (39.2)
Other	6 (2.4)
**Children** (*n*, %^1^)	
Yes	166 (65.1)
No	89 (34.9)
**Economic situation** (*n*, %^1^)	
Very good	51 (20.0)
Good	96 (37.6)
Satisfactory	78 (30.6)
Less good	23 (9.0)
Bad	7 (2.7)
**Number of children** (*n*, %^1^)	
One	62 (24.3)
Two	82 (32.2)
More than two	20 (7.9)
**Hereditary cancer syndrome** (*n*, %^1^)	
HBOC ^a^	196 (76.9)
HNPCC ^b^	33 (12.9)
FAP ^c^	13 (5.1)
Other	13 (5.1)
**Cancer** (*n*, %^1^)	
No	122 (47.8)
Yes ^d^	133 (52.2)
**Prophylactic surgeries** (*n*, %^1^)	
No	119 (46.7)
Yes ^e^	136 (53.3)

Notes. ^a^ Hereditary Breast and Ovarian Cancer. ^b^ Hereditary Non-Polyposis Colorectal Cancer. ^c^ Familial adenomatous polyposis. ^d^ More than one diagnosis was possible, resulting in 66.9% breast cancer, 14.3% ovarian cancer, 3.8% endometrium cancer, 16.5% colon cancer, 16.5% other cancers in those with cancer. ^e^ of those (%^1^): 30.9% ovariectomy, 29.4% mastectomy, 21.3% mastectomy and ovariectomy, 6.6% proctocolectomy, 5.2% hysterectomy, 1.5% other. ^1^ All percentages were calculated using valid cases only, and sums of more than 100% are due to rounding to two decimal places.

**Table 2 ijerph-19-14517-t002:** Results of the binary logistic regression for the desire for children (*n* = 91).

						95% Confidence Interval for Odds Ratio	
	B	SE	Wald	*p*	Odds Ratio	LL ^a^	UL ^b^
Age	0.010	0.061	0.028	0.868	1.010	0.896	1.139
Education	−0.312	0.204	2.351	0.125	0.732	0.491	1.091
Economic situation	−0.197	0.254	0.603	0.437	0.821	0.499	1.351
Relationship status	0.301	0.564	0.285	0.593	1.351	0.448	4.078
Children	0.578	0.596	0.939	0.332	1.782	0.554	5.728
Cancer diagnosis	1.125	0.536	4.395	0.036	3.079	1.076	8.810
Prophylactic surgeries	0.257	0.505	0.259	0.611	1.293	0.481	3.478
Distress	0.079	0.101	0.599	0.439	1.082	0.887	1.320

Notes. The degrees of freedom (df) for the Wald test were all 1. ^a^ Lower limit. ^b^ Upper limit.

**Table 3 ijerph-19-14517-t003:** Results of the binary logistic regression for distress (*n* = 255).

						95 % Confidence Interval for Odds Ratio	
	B	SE	Wald	*p*	Odds Ratio	LL ^a^	UL ^b^
Age	−0.031	0.015	4.356	0.037	0.970	0.942	0.998
Education	0.102	0.096	1.127	0.288	1.107	0.918	1.335
Economic situation	0.366	0.144	6.430	0.011	1.441	1.087	1.912
Relationship status	0.554	0.328	2.862	0.091	1.741	0.916	3.310
Children	−0.013	0.331	0.002	0.968	0.987	0.516	1.888
Cancer diagnosis	−0.532	0.289	3.393	0.065	0.587	0.333	1.035
Prophylactic Surgeries	0.389	0.271	2.059	0.151	1.476	0.867	2.513
Desire for children	0.144	0.359	0.161	0.688	1.155	0.571	2.335

Notes. The degrees of freedom (df) for the Wald test were all 1. ^a^ Lower limit. ^b^ Upper limit.

## Data Availability

The datasets generated during and/or analyzed during the current study are available from the corresponding author on reasonable request.

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
