# Peer review of "Desire for Children and Distress in Women with Hereditary Cancer Syndromes"

_ijerph, 2022, doi:10.3390/ijerph192114517_

Round 1
Reviewer 1 Report (Previous Reviewer 1)
Thank you for addressing previous feedback. No further comments or suggestions.
Author Response
Thanks for your review
Reviewer 2 Report (New Reviewer)
Manuscript deals with a relevant topic with research and clinical implication. nevertheless, some considerations should be given.
Manuscript focused on two specific topics: the distress related to diagnosis and the desire for parenthood. In introduction section, authors explain the state of art for the last one, while the global burden and distress is not adequately explained. Despite commune sense suggest that carriers represents a stressful experience, it is not clear if previous studies focused on this assessment.
In method section authors declared that only women were included in the study. because this was not a inclusive criteria, an explanation of this choice should be given.
in statistical analyses only regression were explained because they were the most relevant analyses. nevertheless, given 4 aims were declared, authors should briefly explain how each aims is tested.
In results, distress is described according to continuous scores (aim 2). nevertheless, because instrument give a cut off, I think that the frequencies of highly distressed respondents has more clinical implication. so i suggest to add this info, also according to subsequent regression analyses, were dicotomic variables is used.
finally, in discussion section, authors stated that "the mutation carriers had increased levels of psychological distress". I think that this is not correct because no comparison is given. Furthermore, also the consideration on risk factors should be given in a less string way, also according that analyses showed only a weak R2.
Author Response
Thank you for the review. See attachment for my answers.

This manuscript is a resubmission of an earlier submission. The following is a list of the peer review reports and author responses from that submission.
Round 1
Reviewer 1 Report
Thank you for making minor revisions and for clarifying my queries where necessary. Please consider the following changes.
1. Consider a change in title given that the aim of your analysis was to examine both desire to have children and distress in women with hereditary cancer.
2. Section 1.1: Please specify hypotheses for each of your aims. What are the factors you hypothesise to influence desire to have children? What are the factors you hypothesise to influence distress? Currently these are included together and this is confusing for the reader.
3. Given that only 18-to-39 years old participants were included in analyses relating to desire to children, can you please clarify in methods the size of this subsample? How many participants were included in analyses exploring desire to have children? How many participants were included in analyses exploring distress? Please specify.
Reviewer 2 Report
Kastner AM et al examined the desire for children in women with hereditary cancer syndromes. Authors investigated the amount of distress and other influencing factors such as age, education, economic status, marital status, presence of children, previous or current cancer and prophylactic surgery. It is an observational study without a specific intervention which confirms general true that most women desire to have children, and of course, the majority of those with hereditary cancer syndromes. Moreover, this study also confirms that women with hereditary cancer syndromes present increased distress and need more support and better family planning. Indeed, there is no new insight through this study in literature. Authors should add new parameters under investigation.